

# Identification of immunogenic cell death gene-related subtypes and risk model predicts prognosis and response to immunotherapy in ovarian cancer

Wenjing Pan[1], Zhaoyang Jia[1], Xibo Zhao[2], Kexin Chang[3], Wei Liu[1] and Wenhua Tan[1]

[1] Department of Gynecology, Second Affiliated Hospital of Harbin Medical University, Harbin, Heilongjiang, China
[2] Department of Gynecological Oncology, Sun Yat-Sen Memorial Hospital of Sun Yat-Sen University, Sun Yat-Sen University of Medical Sciences, Guangzhou, China
[3] Department of Gynecology Oncology, Harbin Medical University Cancer Hospital, Harbin Medical University Cancer Hospital, Harbin, Heilongjiang, China

Corresponding author
Wenhua Tan,
tanwenhua1962@126.com

## ABSTRACT

**Background:** Immunogenic cell death (ICD) has been associated with enhanced anti-tumor immunotherapy by stimulating adaptive immune responses and remodeling the immune microenvironment in tumors. Nevertheless, the role of ICD-related genes in ovarian cancer (OC) and tumor microenvironment remains unexplored.
**Methods:** In this study, high-throughput transcriptomic data from The Cancer Genome Atlas (TCGA) and Gene Expression Omnibus (GEO) databases as training and validation sets separately were obtained and proceeded to explore ICD-related clusters, and an ICD-related risk signature was conducted based on the least absolute shrinkage and selection operator (LASSO) Cox regression model by iteration. Multiple tools including CIBERSORT, ESTIMATE, GSEA, TIDE, and immunohistochemistry were further applied to illustrate the biological roles of ICD-related genes as well as the prognostic capacity of ICD risk signature in OC.
**Results:** Two ICD-related subtypes were identified, with the ICD-high subtype showing more intense immune cell infiltration and higher activities of immune response signaling, along with a favorable prognosis. Additionally, four candidate ICD genes (IFNG, NLRP3, FOXP3, and IL1B) were determined to potentially impact OC prognosis, with an upregulated expression of NLRP3 in OC and metastatic omental tissues. A prognostic model based on these genes was established, which could predict overall survival (OS) and response to immunotherapy for OC patients, with lower-risk patients benefiting more from immunotherapy.
**Conclusion:** Our research conducted a prognostic and prediction of immunotherapy response model based on ICD genes, which could be instrumental in assessing prognosis and assigning immunotherapeutic strategies for OC patients. NLRP3 is a promising target for prognosis in OC.

## INTRODUCTION

Ovarian cancer is the most lethal gynecologic carcinoma (*Achimas-Cadariu, Paun & Pasca, 2023*), with epithelial ovarian cancer (EOC) accounting for approximately 90% of all OC cases (*Romero & Bast, 2012*; *Bowtell et al., 2015*; *Lisio et al., 2019*). The 5-year survival rate remains low, primarily because patients are often diagnosed at advanced stages (III or IV) due to the subtle onset, nonspecific initial symptoms, or limited screening alternatives (*Ali, 2018*). Standard treatment options for EOC are maximal cytoreductive surgery and chemotherapy based on platinum and paclitaxel, which have shown benefits for patients who are sensitive to platinum (*Armstrong et al., 2020*). However, 70% of patients with progressive illness experience a declined condition within 2 years due to gradual loss of sensitivity to chemotherapeutics (*da Costa & Baiocchi, 2021*). There is an urgent need for new biomarkers to predict prognosis and provide effective, low-toxicity treatment. Immunotherapies such as immune-checkpoint inhibitors (*Zamarin et al., 2020*), cancer vaccines (*Tanyi et al., 2018*), and adoptive cell therapy (ACT) (*Kverneland et al., 2020*) have shown promise but lack efficacy in clinical trials due to poor immunogenicity and immune escape of OC (*Reid, Permuth & Sellers, 2017*; *Stewart, Ralyea & Lockwood, 2019*). Enhancing the immunogenicity of ovarian cancer cells is thus key to optimizing immunotherapeutic outcomes.

Immunogenic cell death (ICD) has been proven to be initiated by specific conditions like gamma irradiation, chemotherapy, or immunotherapy, which could kill tumor cells and elicit adaptive immune response; it can also convert non-immunogenic tumor cells into immunogenic ones (*Galluzzi et al., 2020*; *Garg et al., 2015*). This process involves the release of damage-linked molecular patterns (DAMPs), including high-mobility group protein B1 (HMGB1), calreticulin (CRT), and secreted ATP, which could bind to pattern-recognition receptors (PRRs) on dendritic cells (DCs), promoting their maturation (*Galluzzi et al., 2020*; *Fucikova et al., 2020*). Mature DCs recruit effector T cells to the tumor site, inducing differentiation and maturation to secrete interferon (IFN) gamma for tumor eradication (*Zang et al., 2022*; *Bai et al., 2021*). This leads to robust infiltration of myeloid and lymphoid cells, transforming the tumor from "cold" (lacking infiltration of immune cells) to "hot" (with active immune cells). OC is a kind of immunoresponsive malignancy with a complex immune suppression network to counteract anti-tumor immune responses (*Preston et al., 2011*). The tumor microenvironment (TME) is enriched for a large number of immune checkpoint molecules and immunosuppressive cells, collectively creating an immunosuppressive TME that reduces responsiveness to immunotherapy (*Kim et al., 2022*). Thus, ICD has the potential to stimulate anti-tumor immune responses, offering promise for enhanced treatment outcomes in OC patients.

Several ICD modulators have exhibited anti-tumor effects *via* immunogenicity (*Ahmed & Tait, 2020*). For example, studies indicated that IL-1α augmented the effects of cetuximab in head and neck squamous cell carcinoma (*Espinosa-Cotton et al., 2019*). Moreover, certain ICD modulators have proven their capability in various diseases to serve as diagnostic biomarkers. Specifically, HMGB1 is a unique biomarker for several inflammatory skin diseases and a potential novel therapeutic target (*Satoh, 2022*). Given this distinctiveness,

ICD is anticipated to provide innovative insights and modalities in anti-tumor immunotherapy (*Fucikova et al., 2020*). Better identification and evaluation of immunomodulatory therapies may constitute essential means to address the poor prognosis in OC, potentially making ICD an effective strategy (*Stewart, Ralyea & Lockwood, 2019*).

To better understand the importance of ICD in OC, ICD-associated biomarkers were extracted and a risk model was developed, which could aid in deconvolution of the immune microenvironment, prediction of prognosis, and immunotherapeutic response in OC, ultimately appropriate facilitation in clinical decision.

## MATERIALS AND METHODS

### Datasets proceeding

EOC expression data were extracted from The Cancer Genome Atlas (TCGA) and Gene Expression Omnibus (GEO) databases. For the TCGA cohort, read count files from each sample were obtained by the GDC portal tool, and genes with a minimum count of zero in all samples were filtered out, followed by conducting the trimmed mean of M values (TMM) method and calculating the normalized factor supplied by edgeR software. Considering the same histopathology, a large cohort GSE9891 was selected for validation which samples were diagnosed as serous or papillary serous carcinoma, and duplicated genes with the distinct probes were merged by their median values following logarithmic transformation. A total of 362 and 255 samples from TCGA (https://portal.gdc.cancer.gov/) and GSE9891 (https://www.ncbi.nlm.nih.gov/geo/query/acc.cgi?acc=GSE9891) were remained, after excluding samples with unrecorded or less than 30 survival days separately. Moreover, thirty-four ICD genes were derived from a previous study by *Garg, Ruysscher & Agostinis (2016)*.

### Consensus clustering

Specimens from TCGA set as the training cohort were categorized by k-means clustering method based on Euclidean distance among genes performed by the ConsensusClusterPlus package. This process was iterated 1,000 times with 0.8 proportion of resampling to guarantee consistent outcomes. The optimal cluster number (k = 2) ranging from k = 2 to 10 was determined in the elbow and heatmap plots, defined as ICD-high and ICD-low subtypes.

### Screening for differentially expressed genes (DEGs)

The expressed differences of all protein-coding genes between clusters were implemented by the limma package. The filtering thresholds were determined as a corrected *P*-value by Benjamini Hochberg (BH) less than 0.05 and an absolute log2FC value exceeding 1. In addition, the Wilcoxon rank-sum test was also conducted for specific genes or cells from distinct groups.

### Enrichment analysis

Interactions among ICD-related proteins were drawn from the STRING database (*Szklarczyk et al., 2021*) (https://string-db.org/) and customized by Cytoscape. After

conversion from DEG symbols between two ICD subtypes to Entrez IDs applied by org.Hs. eg.db package, clusterProfiler (*Wu et al., 2021*) software was carried out to gain the functional domains of Gene Ontology (GO) from biological process, cellular component, molecular function, and curated pathways of the Kyoto Encyclopedia of Genes and Genomes (KEGG).

## Gene set enrichment analysis

Gene Set Enrichment Analysis (GSEA) offers statistical insights into the biological significance of gene sets (*Liberzon et al., 2015*). To gain a further understanding of the relevant signaling pathways and molecular mechanisms activated in the ICD-high subtype, GSEA was used to analyze the enrichment differences and the enrichment scores (ES) or normalized ES were calculated with curated c2.cp.kegg.v7.4.symbols.gmt datasets chosen as the reference set, which was available from the Molecular Signatures Database (MSigDB, https://www.gsea-msigdb.org/gsea/downloads.jsp).

## Differences in immune characteristics of the two ICD subtypes and correlation analysis

To elucidate the tumor microenvironment (TME) between two ICD subtypes, two estimation methods, as CIBERSORT (https://cibersort.stanford.edu/), ESTIMATE (*Yoshihara et al., 2013*), were applied to deconvolute the bulk transcriptome for estimation of the relative or absolute component of 22 immune cell types, and calculate the stromal or immune scores representing the infiltrated extent of stroma or immune cells in tumor tissues, as well as estimate scores, from which tumor purity could be inferred. Subsequently, comparisons were conducted to compare the association of 22 immune cell types and expression levels of immune checkpoints (ICP) (*Hu et al., 2021*), human leukocyte antigen (HLA) genes (*Ju et al., 2021*), as well as immunomodulators related to tumor-infiltrating lymphocytes, immune cytolytic activity, and interferon response. Finally, the tumor immune dysfunction and exclusion (TIDE, http://tide.dfci.harvard.edu/) algorithm based on the online platform was employed to measure the response of immunotherapy, thus evaluating the interaction among tumor cells, immune cells, and immune escape mechanisms.

## Construction and validation of the prognostic ICD risk signature

First, the prognostic values of ICD cluster-related DEGs were extracted from the univariate Cox regression method. Then least absolute shrinkage and selection operator (LASSO) Cox regression analysis provided in glmnet package was applied performing continuous shrinkage and feature selection to formulate an ICD prognostic risk signature. All differentially expressed ICD genes were considered here, extracting and ranking the optimal lambda values by introducing a penalty term throughout 1,000 iterations, during each of which ten-fold cross-validation was set to derive the best-fit lambda value while minimizing the mean cross-validated error. Finally, the LASSO Cox model was constructed through the formula:

$$Risk\ Score = \sum_{i=1}^{n} k_i * A_i.$$

Here, $A_i$ represents the expression value of the ICD genes, $k_i$ is designated as the regression coefficient, and n denotes the number of optimal solutions derived from the LASSO results. The frequency of genes in each iteration was additionally counted representing their importance. Meanwhile, this ICD risk signature was also validated in the GSE9891 cohort.

## Survival analysis

Kaplan-Meier (KM) analysis and log-rank test were performed using the survminer and survival packages to compare overall survival (OS) between the high- and low-risk groups based on the ICD risk signature in TCGA training and GSE9891 validation cohorts. Univariate Cox regression analysis was adopted to determine potential prognostic capacity among clinical pathological variables and the ICD risk signature, following an examination of prognostic independence by multivariate Cox analysis for OS.

## Human EOC and normal ovarian epithelial tissues

After obtaining approval from the Medical Ethics Committee of the Second Affiliated Hospital of Harbin Medical University (approval number, YJSKY2022-074) and written informed consent from the patients, tissues from 10 women diagnosed as EOC at the Second Affiliated Hospital of Harbin Medical University were retrieved, of whom six cases also manifested omental metastasis and other four cases showed normal omentum. Additionally, six normal ovarian epithelial tissues were obtained. Inclusion criteria were EOC patients without chemotherapy, radiotherapy, or other neoadjuvant therapy; without other malignant tumors; and showing a clear pathological diagnosis. The collected tissues were excised and immobilized in formalin, cut into thin slices, and embedded in paraffin.

## Immunohistochemistry (IHC)

Paraffin sections of ovarian and greater omentum tissues were first dewaxed and hydrated, then subjected to heat-induced antigen retrieval with EDTA (pH = 8.0), and washed five times with distilled water. Sections were blocked with goat serous and incubated with primary antibody (rabbit anti-human NLRP3 antibody (1:500, CY5651; Abways, San Diego, CA, USA)) and murine anti-human IFNG monoclonal antibody (1:1,000, MH45241; Abmart, Shanghai, China) at 4 °C overnight. We used 100 µL of goat anti-rabbit/mouse IgG (Kit-0038; Nakasugi Jinqiao) as a secondary antibody and sections were incubated at room temperature for 30 min. Freshly configured DAB (1:1) was added to stain the tissues, which were then counterstained with hematoxylin. Sections were dehydrated and sealed, and then images were obtained using an Olympus microscope. Three randomly 200× fields-of-view were photographed and preserved. Immunohistochemical semi-quantitative expression analysis was performed using a double-blinded method. A total of 10 ovarian cancer tissues, six normal ovarian epithelial tissues, six metastatic omental, and four normal greater omental cases were analyzed.

ImageJ (NIH, Bethesda, MD, USA) was used to measure integrated optical density (IOD) and area (Area). The average absorbance (AOD) was calculated as AOD = IOD/Area.

## Statistical analysis

Analyses and visualization of high-throughput experimental data were all processed in R software (version 4.3.0) and Linux. Experimental data by IHC were analyzed and plotted by GraphPad Prism 9.0, and the results were expressed as the mean ± standard deviation. All the results were repeated at least three or more times. The t-test or Wilcoxon rank-sum test was used for comparison between the two groups, and the one-way ANOVA or Kruskal–Wallis test was used for comparisons of more than two variables. All statistical studies used $P < 0.05$ or the corrected $P$ value $< 0.05$ as the threshold for statistical significance.

# RESULTS

## Expression patterns and consensus clustering of ICD-related genes in EOC

ICD-related genes in this study were compiled from a large-scale meta-analysis and their expression profiles in EOC samples compared to normal ovarian tissues were first investigated. Elevated expression of ICD genes, including IL10, CALR, IL1B, BAX, FOXP3, IFNG, ATG5, CXCR3, CD8A, CD8B, NLRP3, CASP1, CD4, PRF1, HSP90AA1, IFNB1, PDIA3, CASP8, MYD88, TNF, and IL6, was observed in EOC (Fig. 1A). Then protein-protein interaction (PPI) network was examined and visualized *via* STRING database and CytoScape software, elucidating tight connections among ICD genes (Fig. 1B). A total of 362 eligible OC patients from the TCGA cohort were used as the training cohort to identify potential clusters based on expression patterns of ICD-associated genes. Through the consensus clustering method, two OC-ICD clusters were identified based on Euclidean distance of which cluster C2 exhibited higher expression of ICD-associated genes compared to C1 (Figs. 1C–1E). Hence C2 was labeled as ICD-high subtype and C1 as ICD-low subtype (Fig. 1F). After excluding samples less than 30 survival days, the KM plot revealed different prognoses between the two ICD subtypes (log-rank $P = 0.026$, Fig. 1G). The ICD-high subtype had a better prognosis than the low subtype.

## Exploration of DEGs and functional enrichment analyses between different ICD subtypes

The identification of DEGs and signaling pathways specific to each subtype aided in understanding the molecular mechanisms affecting prognosis. Based on linear fitting and empirical Bayes estimation provided by limma, 1440 DEGs were identified, with 1,220 genes exhibiting higher expression in the ICD-high subtype and 220 genes exhibiting lower (Fig. 2A). GO and KEGG enrichment analyses revealed that overexpressed genes in the ICD-high subtype primarily participated in immunological activities, including lymphocyte-mediated immunity, activation of immune response *via* cell surface receptor signaling pathways, interaction between cytokines and cytokine receptors, and humoral

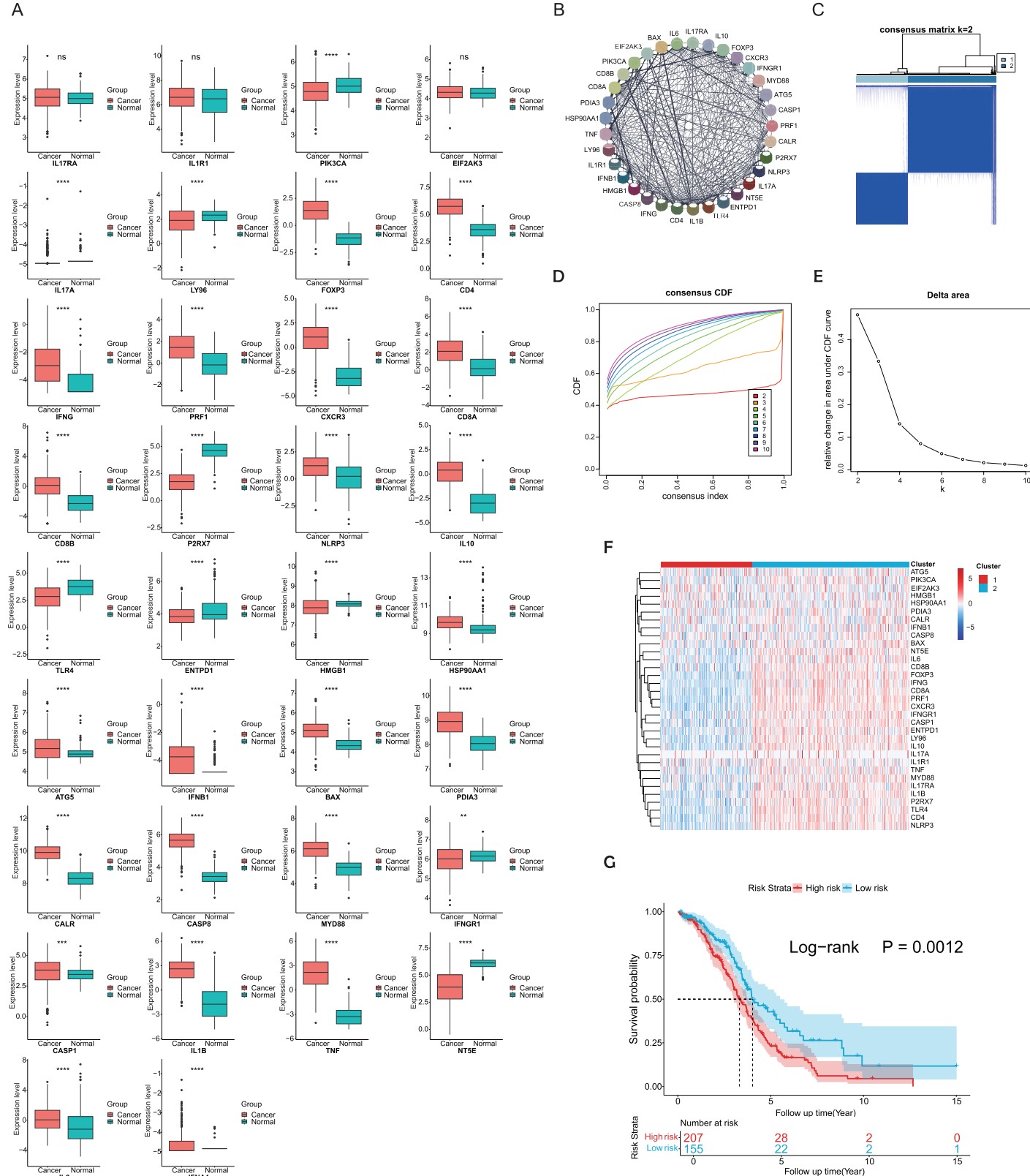

**Figure 1 Expression patterns and consensus clustering of ICD-related genes in EOC.** (A) Box plots showed differential expression of 34 ICD genes between the OC and normal ovary samples. (B) Protein-protein interaction network analysis among ICD-associated genes. (C) The heatmap depicted the consensus clustering of 34 genes in 362 OC samples (k = 2). (D, E) Delta area curves of consensus clusters represented relative variation

**Figure 1 (continued)**
in area under the cumulative distribution function (CDF) curve for k = 2 to 10. (F) The heatmap of ICD-related gene expression in two subtypes. Red indicated high expression and ICD-high subtype while blue indicated low expression and ICD-low subtype. (G) Kaplan-Meier curves of OS in the two ICD-related gene subtypes (log-rank test, $P = 0.026$). ns, no significance; **$P < 0.01$; ***$P < 0.001$ and ****$P < 0.0001$.

immune responses (Figs. 2B, 2C). Furthermore, 17 DEGs were identified as ICD-associated genes with elevated expression in the ICD-high subtype (Fig. 2D). GSEA was utilized to compare the contribution of different gene sets to DEGs between the two subtypes. The results revealed significant activation of several immune-associated signaling pathways in the ICD-high subtype, including the toll-like receptor pathway, Fc gamma R-mediated phagocytosis, the T cell receptor signaling pathway, and leukocyte transendothelial migration (Fig. 2E).

### Characterization of the tumor immune microenvironment landscape in ICD-high and ICD-low subtypes

Growing evidence has indicated that ICD could elicit specific antitumor immune responses. It was further demonstrated substantial activation of immune signaling pathways, particularly in the ICD-high subtype. Analyses of the TME composition here showed the ICD-high subtype had increased stromal, immune, and ESTIMATE scores, and reduced tumor purity compared to the ICD-low subtype ($P < 0.0001$, Fig. 3A). Using the CIBERSORT deconvolution method combined with the LM22 feature matrix, the relative proportions of 22 immune cells were further assessed between the two subtypes in the TCGA training cohort (Fig. 3B). This revealed the ICD-high subtype had notably elevated percentages of M1 and M2 macrophages, resting NK cells, plasma cells, and CD8[+] T cells (Fig. 3C). Correlation analysis revealed strong associations among infiltrating immune cells, demonstrating complexity in OC tissues. For instance, naive B cells, plasma cells, and CD8[+] T cells positively correlated with M1 macrophages, while memory B cells negatively correlated with M0 and M1 macrophages, naive B cells, and plasma cells (Fig. 3D). These infiltration of 22 immune cell types and correlations were further validated using the GSE9891 dataset (Fig. 3E). Furthermore, comparing the expression pattern of the 25 ICP and 17 HLA genes showed virtually upregulated patterns in the ICD-high subtype, which might mean strong immunogenicity in the high subtype ($P < 0.0001$, Figs. 3F, 3G).

### Construction and validation of the prognostic ICD risk signature

The potential capacities of ICD-associated DEGs in predicting prognosis were further explored. Demonstrated by univariate Cox analysis two distinct genes NLRP3 and IFNG out of 17 target ICD genes from all DEGs were identified and showed significant association with OC prognosis (Fig. 4A). Based on machine learning algorithms, the 17 ICD-related DEGs were considered into LASSO models by introducing 1,000 random iterations, each performed by 10-fold cross-validation to identify optimal lambda values and genes, counting the frequency of models in each iteration. The results indicated a model comprised of four distinct genes appearing with the highest frequency (Fig. 4B).

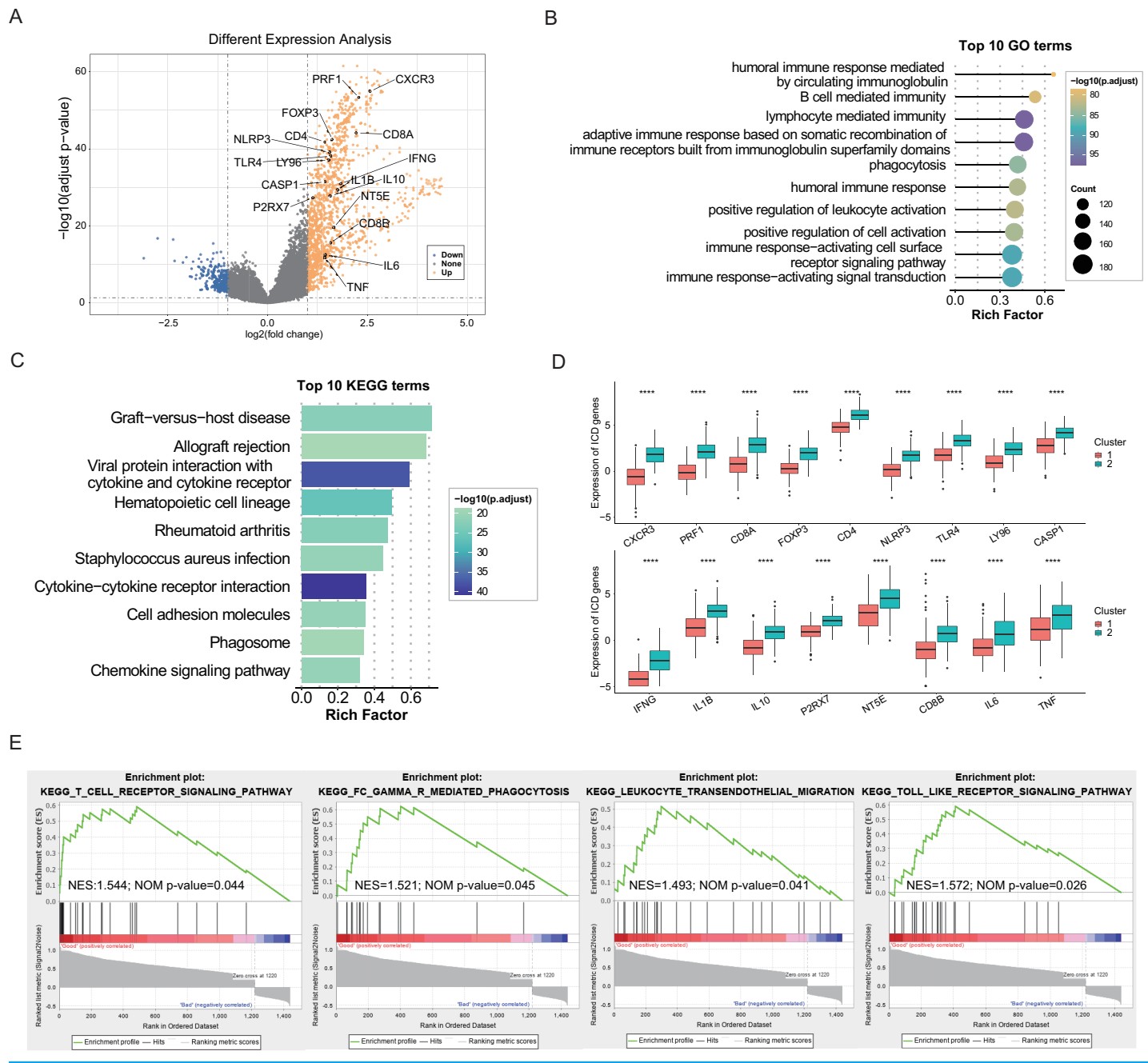

**Figure 2 Exploration of differentially expressed genes (DEGs) and functional enrichment identification in different ICD subtypes.** (A) Volcano plot of DEGs distribution between ICD-high and ICD-low subtypes in the TCGA cohort marked with differentially expressed ICD-related genes. (B, C) GO and KEGG functional enrichment analyses of DEGs among two ICD subtypes. (D) Box plots presenting the differential expression of 17 ICD-related DEGs in two subtypes. (E) GSEA analysis for enriched pathways significantly activated in the ICD-high subtype. ****P < 0.0001.

Components of these four-gene models were further examined conforming to a consistent composition (Figs. 4C, 4D). Specifically, the ranking frequency of IFNG, NLRP3, FOXP3, and IL1B was settled after 1,000 iterations, suggesting sequential significance for prognosis among these ICD genes (Fig. 4E). A risk score model was thus developed based on these

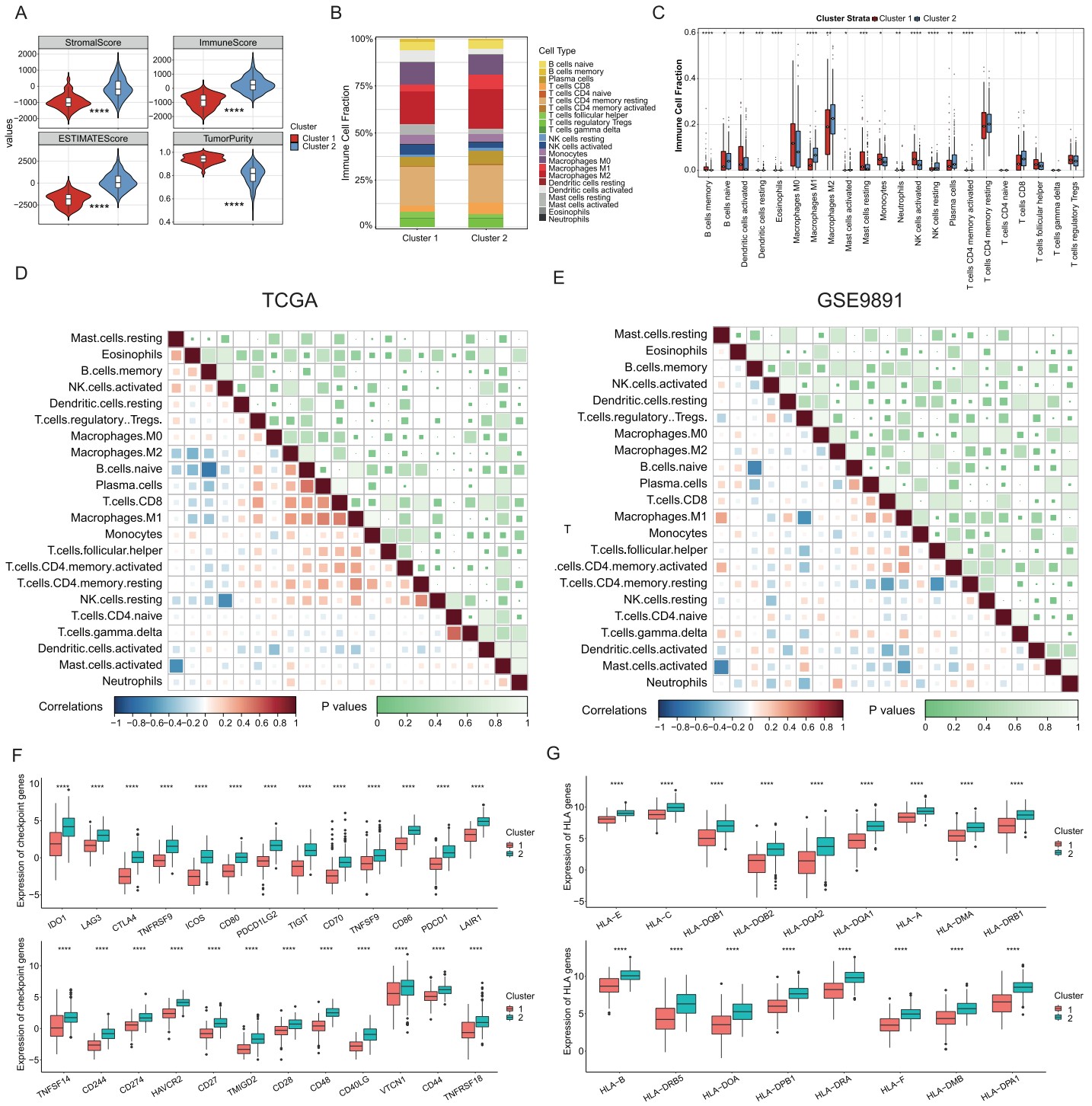

**Figure 3  Analyses of tumor immune microenvironment landscape in ICD-high and ICD-low subtypes.** (A) Violin plots showed the estimation of the stromal score, immune score, ESTIMATE score, and tumor purity between the ICD-high and ICD-low subtypes. (B) Relative percentages of immune infiltration of different immune cell types in ICD-high and ICD-low subtypes, assessed by CIBERSORT. (C) Box plots showed significant differences in 22 immune cell types between ICD-high and -low subtypes. (D, E) Correlation analysis of 22 immune cell species from the TCGA (D) and GSE9891 (E) cohorts, using Pearson's algorithm. (F, G) Box plots showed differential expression of multiple ICP (F) and HLA (G) genes between ICD-high and -low subtypes. *$P < 0.05$; **$P < 0.01$; ***$P < 0.001$ and ****$P < 0.0001$.

genes, with coefficients derived from the LASSO algorithm, fitting as risk score = −0.02139749 * FOXP3 − 0.30089077 * IFNG + 0.05682286 * NLRP3 + 0.02748733 * IL1B. Correlations between risk score and survival status were investigated in the TCGA training cohorts, of which two risk groups were categorized based on the median risk value, and results showed patients with higher risk scores preferred more deaths and shorter survival time (Fig. 4F). Furthermore, a significant trend also revealed poorer OS in the high-risk group compared to the low-risk group (log-rank $P = 0.0012$, Fig. 4G), as well as in the GSE9891 validation cohort (log-rank $P = 0.014$, Fig. 4H).

Univariate and multivariate Cox regression analyses were utilized to assess the predictive performance of the ICD risk model compared with other clinical signatures. Both tumor remnant and the risk model were identified as significant prognostic indicators associated with survival, as a high ICD risk score was indicated as a risk factor ($P = 0.002$, Fig. 4I). Multivariate Cox assessment showed that the ICD risk score was an independent predictor among other clinical parameters ($P = 0.005$, Fig. 4J).

## Exploration of the association between ICD risk signature and tumor immunotherapy

Since the significance of ICD in anti-tumor immune response has been acknowledged, the association between the ICD risk score and the tumor immunotherapy was examined. Results indicated a link between the two indices, with higher risk scores negatively correlated with CD8 cells, follicular helper T cells, and M1 macrophages in the TCGA cohort ($P < 0.05$, Fig. 5A), as evidenced in the GSE9891 cohort ($P < 0.05$, Fig. 5B).

The TIDE score was utilized to evaluate the clinical efficacy of immunotherapy across different ICD risk-score groups. As the TIDE score increased, susceptibility to immune evasion was also escalated, suggesting a potential reduced benefit from immune-checkpoint inhibitor (ICI) treatment. Results revealed a higher TIDE score in the high-risk group compared to the low-risk group ($P < 0.001$, Fig. 5C), whereas the microsatellite instability (MSI) score and the T-cell dysfunction score were higher in the low-risk group ($P < 0.0001$, Figs. 5D, 5E), against the higher T-cell exclusion score in the high-risk group ($P < 0.0001$, Fig. 5F). Moreover, excluding samples with less than one survival month, there was no significant difference in risk scores between the immunotherapeutic non-responder and responder groups (Fig. 5G); however, samples in the immunotherapeutic non-response group proved higher ICD risk scores after excluding samples with less than three survival months ($P < 0.05$, Fig. 5H), suggesting that immunotherapy might be more beneficial for the low-risk group.

## Validation for expression of ICD-related genes in clinical tissues

Distinct expression patterns of specific ICD genes were verified by immunohistochemical analyses in clinical tissues. NLRP3 and IFNG, the top two genes in the risk model, were selected and statistically elevated in OC tissues compared to normal ovarian tissues ($P < 0.05$). Both genes also showed an increasing tendency in metastatic *vs.* non-metastatic omental tissues, with a statistically significant difference in NLRP3 expression (Figs. 6A–6D). The experiment results confirmed the upregulation of NLRP3 in OC and

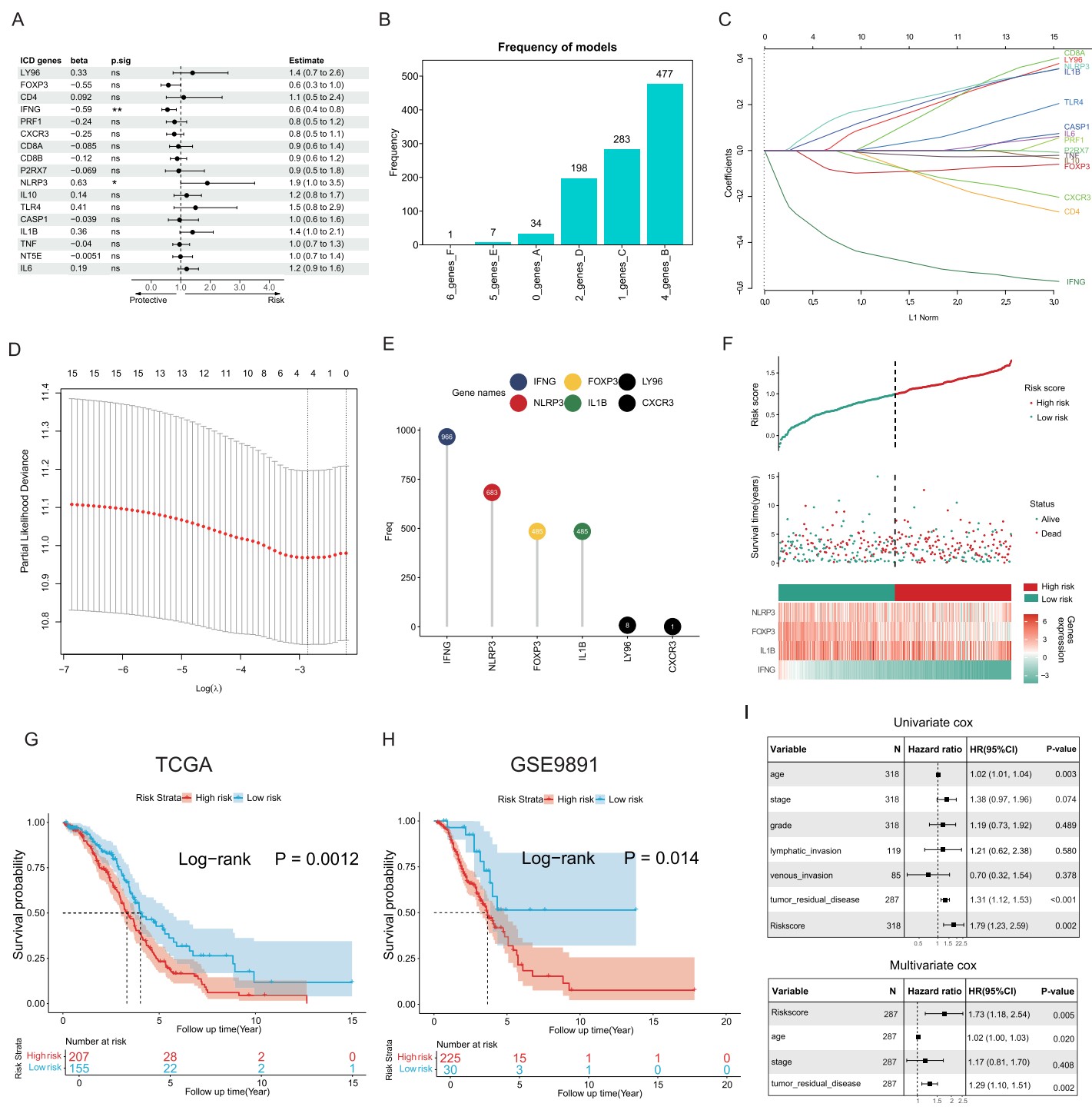

**Figure 4 Construction and validation of the prognostic ICD risk signature.** (A) Univariate Cox analysis to assess the prognostic value of ICD-related differential genes in terms of OS. (B) The feature set consisting of four genes appeared highest frequency after iterations. (C–E) LASSO Cox analysis identified 4 ICD-related genes associated with OS in the TCGA cohort. (F) Ranking risk score, survival status distribution, and heatmap of the prognostic gene signature of each OC patient in the TCGA database. (G, H) Kaplan-Meier curves evaluated the prognostic significance of the risk model in the TCGA (G) and GSE9891 (H) cohorts. (I, J) Univariate and multivariate Cox analyses were performed to determine the relationship and independence in the prognosis of ICD risk signature in patients with OC. *P < 0.05 and **P < 0.01.

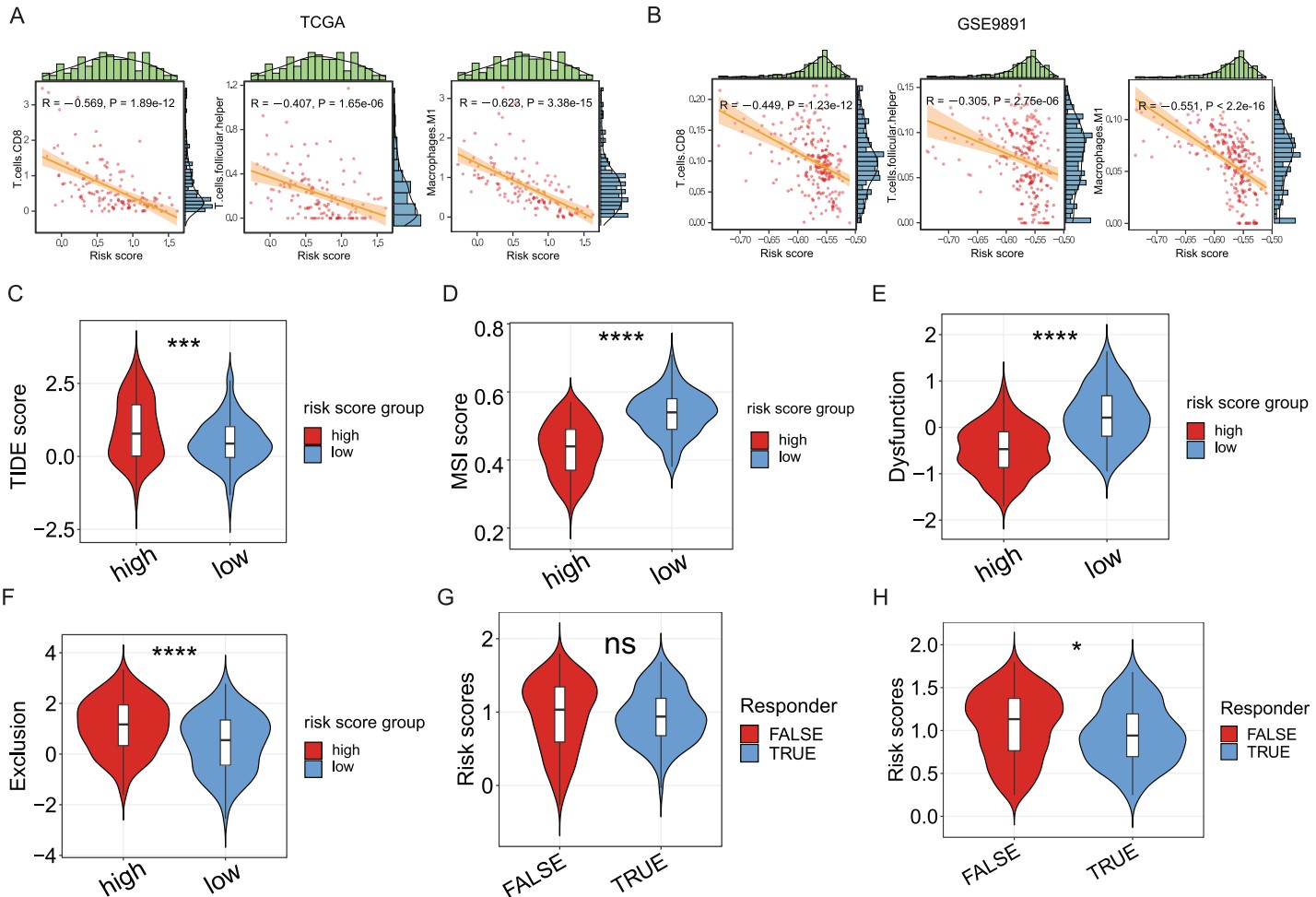

**Figure 5 The association between ICD risk signature and the tumor immune microenvironment and immunotherapy.** (A, B) The scatterplots showed the high relevance between the risk score and the infiltration of CD8 cells, M1 macrophages, and follicular helper T cells (A), which was further validated in the GSE9891 cohort (B). (C–F) Violin plots visualized the TIDE (C), MSI (D), and T-cell dysfunction (E) and exclusion (F) scores between the high- and low-risk groups; (G, H) Violin plots depicted the correlation of ICD risk score and response to immunotherapy, after excluding samples with a prognosis of less than 1 month (G) or 3 months (H). ns, no significance; *P < 0.05; ***P < 0.001 and ****P < 0.0001.

metastatic omentum, enhancing understanding of the related characterization of ICD genes involved in OC progression.

## DISCUSSION

Malignant tumors are the leading cause of death worldwide (*Timar & Kashofer, 2020*). Although low incidence, the mortality rate of OC is higher than that of endometrial or cervical cancers (*Stewart, Ralyea & Lockwood, 2019*; *Kuroki & Guntupalli, 2020*), and has affected younger women (*Lheureux et al., 2019*). Immunotherapy represents the most promising therapeutic option for incurable cancers such as ovarian cancer (*Marth et al., 2019*). However, fewer patients would derive benefit from these treatments due to high heterogeneity in TMEs, low immunogenicity, and immune-escape mechanisms of OC cells. Recent studies showed that ICD could trigger anticancer immune responses (*Ahmed*

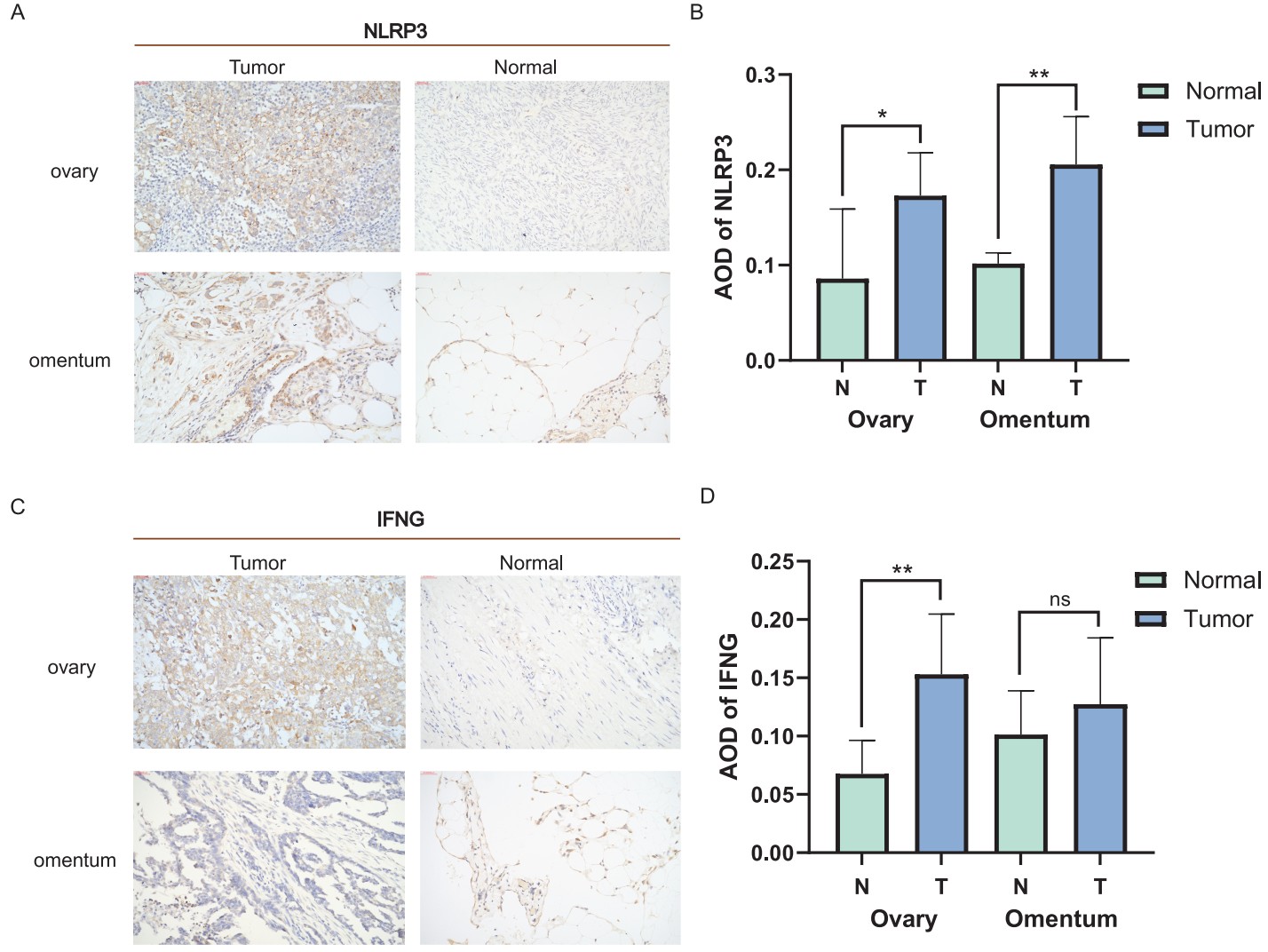

**Figure 6 Expression of ICD-related genes in clinical tissues.** (A, B) Immunohistochemical staining was used to verify the expression pattern of NLRP3 in normal and malignant tissues of the ovary and omentum (A) and the AOD results (B); (C, D) immunohistochemical results of IFNG in normal and malignant tissues of ovary and omentum (C) and the AOD results (D). ns, no significance; *$P < 0.05$ and **$P < 0.01$.

& Tait, 2020; Messmer, Snyder & Oberst, 2019) with direct elimination of cancer cells through augmenting anti-tumor immunity, resulting in an extended effect of anti-cancer medication (Kroemer et al., 2013). One clinical trial showed that chemotherapy-induced ICD could transform "cold" tumors into "hot" tumors (Li et al., 2021). While numerous studies have underscored the importance of ICD in innate immunity and antitumor effects (Zhu et al., 2021; Jin & Wang, 2021), most research on ICD has focused on individual ICD-related genes or a single immune cell type. The characterization of OC mediated by the concerted action of ICD remains incomplete.

This research established a robust correlation between ICD-related genes and OC prognosis as well as the TME. Clustering OC into two subtypes based on the expression of 34 ICD-related genes, the results indicated the ICD-high subtype had a superior prognosis

to the ICD-low subtype. Analyses of the DEGs and signaling pathways between two ICD subtypes revealed 1,440 differentially regulated genes, with 1,220 up-regulated, particularly in the ICD-high subtype. GO and KEGG functional analyses indicated that upregulated DEGs were involved in lymphocyte-mediated immunity, cytokine-cytokine receptor interactions, and immune response-activating cell surface receptor signaling pathways. GSEA confirmed marked activation of immune-associated signaling pathways in the ICD-high subtype, including the T-cell receptor signaling pathway, phagocytosis mediated by FcγR, *etc*. Of the DEGs, 17 were ICD-associated genes, including CXCR3, CD8A, IFNG, IL1B, IL10, NT5E, CD8B, IL6, TNF, PRF1, FOXP3, CD4, NLRP3, TLR4, LY96, CASP1, and P2RX7, all of which were up-regulated in ICD-high subtype. These findings suggested a strong linkage in favorable prognosis in the ICD-high subtype with immune system modulation and stimulation.

The immune system serves as the primary defense against both external and internal threats (*Lakins et al., 2018*) and is primarily composed of innate and adaptive immune cells (*Chu et al., 2022*). The TME, which is characterized as a dynamically shifting and intricate network, is pivotal in tumor development, progression, and metastasis (*Wei et al., 2020*). The highly heterogeneous nature of OC is primarily reflected in its complex TME, which exhibits a distinct immunosuppressive phenotype (*Drakes & Stiff, 2018*), and prior research has indicated this heterogeneity may directly contribute to treatment failure (*McPherson et al., 2016*). Consistently, our research also investigated the disparities in immune-cell infiltration between the two ICD subtypes. In the TCGA cohort, we observed infiltration of M1 macrophages, plasma cells, and CD8$^+$ T-cells in the ICD-high subtype, which was associated with a better prognosis. This was validated in the GSE9891 cohort, though results from these two datasets were not identical, highlighting the complexity of the TME in OC. Nevertheless, immune cell infiltration was higher in the ICD-high subtype across both cohorts. Our findings are consistent with previous studies indicating that CD8$^+$ T cell markers are strongly associated with favorable prognosis (*Bindea et al., 2013*) and that M1 macrophages exhibit an inhibitory effect on tumor cell growth (*Qian & Pollard, 2010*). Furthermore, significant correlations were observed between these immune cell types, suggesting their interactions might synergistically regulate tumor progression. Furthermore, the results of the ICP and HLA genetic pattern showed that the ICD-high subtype was associated with an immune hot phenotype, while the ICD-low subtype had the opposite association.

Subsequently, a prognostic risk-score model was developed and validated using four ICD-related genes (IFNG, NLRP3, FOXP3, and IL1B), which classified OC patients into high- and low-risk groups. Similar to previous studies (*Wang et al., 2021*; *Zhou et al., 2023*), this risk-score model exhibited significant predictive capability regarding OS and showed potential as a standalone prognostic marker. The efficacy of the ICD model in evaluating the advantages of immunotherapy was assessed using the TIDE score, revealing individuals in the high-risk group showed elevated TIDE scores, which suggested an increased susceptibility to immune evasion. Furthermore, an increase in the TIDE predictive score corresponded to a poorer prognosis. Therefore, individuals in the low-risk group were anticipated to reflect a more favorable prognosis compared with those in the

high-risk group. This observation was congruent with our prior findings (Figs. 4G, 4H). When assessing the relationship between immunotherapeutic response and ICD risk score, no notable linkage was observed after excluding cases with survival shorter than 1 month; however, setting the threshold at 3 months, the ICD risk score was significantly increased in the immunotherapeutic non-response group, possibly associated with the lack of immunotherapy with shorter survival of previous ones. The high-risk group exhibited reduced T-cell dysfunction and elevated T-cell rejection scores compared with the low-risk group, suggesting the lower immunotherapeutic response rate was due to immune evasion with both of these two modalities (*Jiang et al., 2018*). On the other hand, the low-risk group showed higher MSI scores. Previous studies suggested that a high mutational load resulting from MSI could enhance tumor immunogenicity and sensitivity to anti-PD1 treatment (*Mandal et al., 2019*). It is plausible that the low-risk group (with presumably lower immune evasion) benefited more from immunotherapy. Our findings collectively indicated that ICD risk scores could be used to evaluate the effect of immunotherapy and the prognosis of OC.

This study might still have some limitations, including its retrospective design which might lead to selection bias. Prospective clinical studies could help validate the prognostic accuracy of the ICD model. Our study discovered that four ICD genes might influence OC prognosis, and confirmed high expression of NLRP3 and IFNG in OC tissues, as well as NLRP3 in metastatic omental tissues by IHC. However, their specific mechanisms need further investigation *in vitro* and *in vivo*, and more experiments are ongoing to explore their biological roles in OC. Once validated, they could potentially serve as therapeutic targets for OC.

## CONCLUSIONS

In summary, our study identified two ICD subtypes of OC and examined their association with TME, specifically infiltration of immune cells. A prognostic model based on ICD-related genes was also developed to predict overall survival and immunotherapeutic response. This research offers a strategy for assessing and intervening in the OC prognosis and treatment. Further research could integrate this model with clinical factors and targeted drug screening to enhance its practicality and accuracy in evaluating OC immunotherapy.

## ACKNOWLEDGEMENTS

We would like to thank Dr. Linyao Zheng and Dr. Tianyu Ma for their valuable assistance in data analysis and statistics. We thank LetPub for its linguistic assistance during the preparation of this manuscript.

### Funding

The authors received no funding for this work.
## Competing Interests

The authors declare that they have no competing interests.

## Author Contributions

- Wenjing Pan conceived and designed the experiments, performed the experiments, analyzed the data, prepared figures and/or tables, and approved the final draft.
- Zhaoyang Jia performed the experiments, prepared figures and/or tables, authored or reviewed drafts of the article, and approved the final draft.
- Xibo Zhao conceived and designed the experiments, performed the experiments, analyzed the data, authored or reviewed drafts of the article, and approved the final draft.
- Kexin Chang performed the experiments, analyzed the data, prepared figures and/or tables, and approved the final draft.
- Wei Liu performed the experiments, authored or reviewed drafts of the article, and approved the final draft.
- Wenhua Tan conceived and designed the experiments, authored or reviewed drafts of the article, and approved the final draft.

## Human Ethics

The following information was supplied relating to ethical approvals (*i.e.*, approving body and any reference numbers):

Medical Ethics Committee of the Second Affiliated Hospital of Harbin Medical University.

## Data Availability

The data is available at NCBI GEO: GSE9891 and National Cancer Institute, GDC Data Portal: TCGA-OV.

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
