# Peer review of "Identification of immunogenic cell death gene-related subtypes and risk model predicts prognosis and response to immunotherapy in ovarian cancer"

_PeerJ, doi:10.7717/peerj.18690_

## Round 0.1 · original submission · Major Revisions

The article's content is quite substantial, but both the language and format require significant improvement. In addition to the reviewers' comments, two major issues stand out:

1. The paper excessively uses first-person pronouns, with "we" appearing 69 times and "ours" 24 times. This is inappropriate for a scientific paper, which is typically written in a more formal, impersonal style.
2. The quality of the figures is inadequate. On an A4 layout, some figures appear noticeably blurry, and the design of the composite sub-figures is poorly executed.

Both issues should be addressed to enhance the overall quality of the manuscript.

When revising your manuscript, please consider all issues mentioned in the comments from the reviewers and the editor carefully and provide suitable responses for any comments. Please note that your revised submission may need to be re-reviewed.

·

Basic reporting

The introduction is well-written but could benefit from a deeper discussion on the significance of ICD in the context of ovarian cancer specifically. The connection between ICD and the immune microenvironment in ovarian cancer should be elaborated further to establish the study’s relevance.

The abstract provides a good summary but lacks specific details on methods and key findings. Including brief data points or results could strengthen the abstract and provide a clearer overview of the study's impact.

Experimental design

The methods section lacks sufficient detail on the analytical tools and criteria used for subtype classification and DEG identification. Please provide more information on the specific thresholds, software, or algorithms used.
Clarify how the cohort was selected and whether any exclusion criteria were applied. Additionally, the rationale behind choosing certain tools like CIBERSORT and GSEA should be briefly discussed to justify their appropriateness for this study.

The discussion should address potential limitations of the study, such as the generalizability of the findings to broader populations or the limitations of the chosen computational methods. This is crucial for the credibility and future application of the findings.
Further discussion on how this study compares with existing research on ICD in ovarian cancer or other cancers would be beneficial. Highlight any novel contributions or discrepancies with past studies.

Validity of the findings

The results are promising but require more quantitative data to support the conclusions. For instance, include specific survival statistics, hazard ratios, or p-values where applicable.

The characterization of the two ICD-associated subtypes needs to be more comprehensive. Provide more in-depth analysis and visualization of the immune cell infiltration data, and explain how these subtypes differ significantly in terms of their molecular profiles.

Figures and tables should be carefully reviewed for clarity and completeness. Ensure that all are appropriately labeled and include relevant statistical annotations. For instance, survival curves should include confidence intervals or p-values to convey statistical significance.

The conclusion section is well-formulated but could be more impactful by explicitly stating the clinical implications of the prognostic model and how it might be integrated into current ovarian cancer treatment strategies. Additionally, suggest potential avenues for future research that could build on your findings.

Additional comments

The title is clear but could be more specific. Consider revising to explicitly mention "ICD-related genes" and "prognostic model" to better reflect the study's contributions.

The manuscript presents valuable insights into the role of ICD-related genes in ovarian cancer and their potential in prognostic modeling and immunotherapy response prediction. However, the manuscript needs revisions in methodological detail, data presentation, and discussion of limitations to strengthen the validity and applicability of the findings.

·

Basic reporting

In this study, the authors used high-throughput experimental data and multiple analytical tools to investigate the role of ICD-related genes in ovarian cancer (OC). The study identified two ICD-associated subtypes, and the authors also identified differentially expressed genes (DEGs) between these two subtypes. Based on these ICD-related DEGs, a prognostic model has been developed to predict overall survival in OC patients and correlate with immunotherapy response.

The general idea of the work is worth studying and relevant to the current field of application. The authors have nicely related their findings with literature in the introduction and discussion. However, the manuscript lacks novelty and clarity in terms of the approaches used for findings. I do have a few comments/suggestions that, I believe, if incorporated in the manuscript will enhance the scientific rigor of the manuscript. A detailed list of comments can be found below.

Experimental design

NA

Validity of the findings

List of questions/comments

1) For prediction, the authors are using LASSO for variable selection and then multivariate cox regression is used for prediction. What are the training or discovery and testing or validation cohort that have been used? Without validation of the signatures in additional independent cohort, the training data based model performance is not acceptable since the good performance could be due to over-fitting.


5) Please provide validation of all the major findings using additional independent cohort. Novel findings without validation are not acceptable.

Additional comments

1) Please detail the data pre-processing and normalization procedure.

2) Are the authors fitting univariate cox model to the entire dataset or there are some prior knowledge about the gene sets?

3) Did the authors use cross-validation in the variable selection or prediction model building? Please explain the detailed algorithm used.

4) For statistical analysis including DEGs, please correct for multiplicity using either Benjamini Hochberg (BH) or "BY" method and report the adjusted p-values. The marginal p-values may give false positive findings.

5) There are several grammatical errors throughout the manuscript and several sentences are even incomplete which hinders the reading flow. The authors might consider getting it reviewed by a professional writer.

---

## Round 0.2 · Minor Revisions

Please explain how this paper is different from these articles (not cited):

https://peerj.com/articles/18235/ Liu Z, Luo Y, Su L, Hu X. Identification of immunogenic cell death-related subtypes used for predicting survival and immunotherapy of endometrial carcinoma through a bioinformatics analysis. Medicine (Baltimore). 2023 Aug 4;102(31):e34571. doi: 10.1097/MD.0000000000034571.

Zhang W, Liu T, Jiang L, Chen J, Li Q, Wang J. Immunogenic cell death-related gene landscape predicts the overall survival and immune infiltration status of ovarian cancer. Front Genet. 2022 Nov 8;13:1001239. doi: 10.3389/fgene.2022.1001239.

There may be others - please check the literature, and cite appropriate literature.

·

Basic reporting

Satisfactory.

Experimental design

NA.

Validity of the findings

Authors have validated their findings

Additional comments

The authors have successfully addressed all the questions raised in the review. I have no further comments/questions. I recommend the article for publication.

---

## Round 0.3 · accepted · Accept

While I am recommending acceptance, the paper would have been improved by including relevant citations and how your study is different rather than a rebuttal to me.